# DATA-EFFICIENT MUTUAL INFORMATION NEURAL ESTIMATOR FOR STATISTICAL DEPENDENCY TESTING

## ABSTRACT

Measuring Mutual Information (MI) between high-dimensional, continuous, random variables from observed samples has wide theoretical and practical applications. Recent works have developed accurate MI estimators through provably low-bias approximations and tight variational lower bounds assuming abundant supply of samples, but require an unrealistic number of samples to guarantee statistical significance of the estimation. In this work, we focus on improving data efficiency and propose a Data-Efficient MINE Estimator (DEMINE) that can provide a tight lower confident interval of MI under limited data, through adding cross-validation to the *MINE* lower bound (Belghazi et al., 2018). Hyperparameter search is employed and a novel meta-learning approach with task augmentation is developed to increase robustness to hyperparameters, reduce overfitting and improve accuracy. With improved data-efficiency, our DEMINE estimator enables statistical testing of dependency at practical dataset sizes. We demonstrate the effectiveness of DEMINE on synthetic benchmarks and real world fMRI data, with application of inter-subject correlation analysis.

## 1 INTRODUCTION

Mutual Information (MI) is an important, theoretically grounded measure of similarity between random variables. MI captures general, non-linear, statistical dependencies between random variables. MI estimators that estimate MI from samples are important tools widely used in not only subjects such as physics and neuroscience, but also machine learning ranging from feature selection and representation learning to explaining decisions and analyzing generalization of neural networks.

Existing studies on MI estimation between general random variables focus on deriving asymptotic lower bounds and approximations to MI under infinite data, and techniques for reducing estimator bias such as bias correction, improved signal modeling with neural networks and tighter lower bounds. Widely used approaches include the k-NN-based KSG estimator (Kraskov et al., 2004) and the variational lower-bound-based Mutual Information Neural Estimator (*MINE*) family (Belghazi et al., 2018; Poole et al., 2018).

Despite the empirical and asymptotic bias improvements, MI estimation has not seen wide adoption. The challenges are two-fold. First, the analysis of dependencies among variables - let alone any MI analyses for scientific studies - requires not only an MI estimate, but also confidence intervals (Holmes & Nemenman, 2019) around the estimate to quantify uncertainty and statistical significance. Existing MI estimators, however, do not provide confidence intervals. As low probability events may still carry a significant amount of information, the MI estimates could vary greatly given additional observations (Poole et al., 2018). Towards providing upper and lower bounds of true MI under limited number of observations, existing MI lower bound techniques assume infinite data and would need further relaxations when a limited number of observations are provided. Closest to our work, Belghazi et al. (2018) studied the lower bound of the *MINE* estimator under limited data, but it involves bounds on generalization error of the signal model and would not yield useful confidence intervals for realistic datasets. Second, practical MI estimators should be insensitive to the choice of hyperparameters. An estimator should return a single MI estimate with its confidence interval irrespective of the type of the data and the number of observations. For learning-based approaches, this means that the model design and optimization hyperparameters need to not only be determined automatically but also taken into account when computing the confidence interval.

Towards addressing these challenges, our estimator, DEMINE, introduces a predictive MI lower bound for limited samples that enables statistical dependency testing under practical dataset sizes. Our estimator builds on top of the *MINE* estimator family, but performs cross-validation to remove the need to bound generalization error. This yields a much tighter lower bound agnostic to hyperparameter search. We automatically selected hyperparameters through hyperparameter search, and a new cross-validation meta-learning approach is developed, based upon few-shot meta-learning, to automatically decide initialization of model parameters. Meta-overfitting is strongly controlled through task augmentation, a new task generation approach for meta-learning. With these improvements, we show that DEMINE enables practical statistical testing of dependency for not only synthetic datasets but also for real world functional Magnetic Resonance Imaging (fMRI) data analysis capturing nonlinear and higher-order brain-to-brain coupling.

Our contributions are summarized as follows: 1) A data-efficient Mutual Information Neural Estimator (DEMINE) for statistical dependency testing; 2) A new formulation of meta-learning using Task Augmentation (Meta-DEMINE); 3) Application to real life, data-scarce applications (fMRI).

## 2 RELATED WORK

### 2.1 MI ESTIMATION

A widely used approach for estimating MI from samples is using k-NN estimates, notably the KSG estimator (Kraskov et al., 2004). Gao et al. (2017) provided a comprehensive review and studied the consistency and of asymptotic confidence bound of the KSG estimator (Gao et al., 2018). MI estimation can also be achieved by estimating individual entropy terms through kernel density estimation (Ahmad & Lin, 1976) or cross-entropy (McAllester & Statos, 2018). Despite their good performance on random variables with few dimensions, MI estimation on high-dimensional random variables remains challenging for commonly used Gaussian kernels. Fundamentally, estimating MI requires accurately modeling the random variables, where high-capacity neural networks have shown excellent performance on complex high-dimensional signals such as text, image and audio.

Recent works on MI estimation have focused on developing tight asymptotic variational MI lower bounds where neural networks are used for signal modeling. The IM algorithm (Agakov, 2004) introduces a variational MI lower bound, where a neural network $q(z|x)$ is learned as a variational approximation to the conditional distribution $P(Z|X)$. The IM algorithm requires the entropy, $H(Z)$, and $E_{XZ} \log q(z|x)$ to be tractable, which applies to latent codes of Variational Autoencoders (VAEs) and Generative Adversarial Networks (GANs) as well as categorical variables. Belghazi et al. (2018) introduces MI lower bounds *MINE* and *MINE-f* which allow the modeling of general random variables and shows improved accuracy for high-dimensional random variables, with application to improving generative models. Poole et al. (2018) introduces a spectrum of energy-based MI estimators based on *MINE* and *MINE-f* lower bounds and a new TCPC estimator inspired by Contrastive Predictive Coding (Oord et al., 2018) for the case when multiple samples from $P(Z|X)$ can be drawn.

Our work introduces cross-validation to the *MINE-f* estimator. We derive the lower bound of *MINE-f* under limited number of samples, and introduce meta-learning and hyperparameter search to enable practical statistical dependency testing.

### 2.2 GENERAL STATISTICAL DEPENDENCY TESTING

Existing works in general statistical dependency testing (Bach & Jordan, 2002; Gretton et al., 2005a; Berrett & Samworth, 2019) have developed non-parametric independent criterions based on correlation and mutual information estimators equivalent to testing $I(X; Z) = 0$, followed by detailed bias and variance analyses. Our approach for independent testing suggest a different direction by harnessing the generalization power of neural networks and may improve test performance on complex signals. The p-values provided by our test do not involve approximated distributions and hold for small number of examples and arbitrary number of signal dimensions. As different statistical dependency testing approaches have explicit or implicit assumptions and biases that make them suitable in different situations, a fair comparison across different approaches is a challenging task. Instead, we focus on a self-contained presentation of our dependency test, and provide preliminary comparisons with a widely studied Hilbert-Schmidt independence criterion (HSIC) (Gretton et al., 2005a) in the appendix.

## 2.3 META LEARNING

Meta-learning, or "learning to learn", seeks to improve the generalization capability of neural networks by searching for better hyperparameters (Maclaurin et al., 2015), network architectures (Pham et al., 2018), initialization (Finn et al., 2017a; 2018; Kim et al., 2018) and distance metrics (Vinyals et al., 2016; Snell et al., 2017). Meta-learning approaches have shown significant performance improvements in applications such as automatic neural architecture search (Pham et al., 2018), few-shot image recognition (Finn et al., 2017a) and imitation learning (Finn et al., 2017b).

In particular, our estimator benefits from the Model-Agnostic Meta-Learning (MAML) (Finn et al., 2017a) framework which is designed to improve few-shot learning performance. A network initialization is learned to maximize its performance when fine-tuned on few-shot learning tasks. Applications include few-shot image classification and navigation.

We leverage the model-agnostic nature of MAML for MI estimation between generic random variable and adopt MAML for maximizing MI lower bounds. To construct a collection of diverse tasks for MAML learning from limited samples, inspired by MI's invariance to invertible transformations, we propose a task-augmentation protocol to automatically construct tasks by sampling random transformations to transform the samples. Results show reduced overfitting and improved generalization.

## 3 BACKGROUND

In this section, we will provide the background necessary to understand our approach[1]. We define $X$ and $Z$ to be two random variables, $P(X, Z)$ is the joint distribution, and $P(X)$ and $P(Z)$ are the marginal distributions over $X$ and $Z$ respectively. Our goal is to estimate MI, $I(X; Z)$ given independent and identically distributed (*i.i.d.*) sample pairs $(x_i, z_i)$, $i = 1, 2 \ldots n$ from $P(X, Z)$. Let $\mathcal{F} = \{T_\theta(x, z)\}_{\theta \in \Theta}$ be a class of scalar functions, where $\theta$ is the set of model parameters. Let $q(x|z) = p(x) \frac{e^{T_\theta(x,z)}}{\mathbb{E}_{(x,z) \sim P_{XZ}} e^{T_\theta(x,z)}}$. Results from previous works (Belghazi et al., 2018; Poole et al., 2018) show that the following energy-based family of lower bounds of MI hold for any $\theta$:

$$
\begin{aligned}
I(X; Z) &\geq \mathbb{E}_{(x,z) \sim P_{XZ}} \log \frac{q(x|z)}{p(x)} = \mathbb{E}_{(x,z) \sim P_{XZ}} T_\theta(x, z) - \mathbb{E}_{x \sim P_X} \log \mathbb{E}_{z \sim P_Z} e^{T_\theta(x,z)} \triangleq I_{\text{EB1}} \\
&\geq \mathbb{E}_{(x,z) \sim P_{XZ}} T_\theta(x, z) - \log \mathbb{E}_{x \sim P_X, z \sim P_Z} e^{T_\theta(x,z)} \triangleq I_{\text{MINE}} \\
&\geq \mathbb{E}_{(x,z) \sim P_{XZ}} T_\theta(x, z) - \mathbb{E}_{x \sim P_X, z \sim P_Z} e^{T_\theta(x,z)} + 1 \triangleq I_{\text{MINE-f}}, I_{\text{EB}}
\end{aligned}
\tag{1}
$$

where, $\mathbb{E}$ is the expectation over the given distribution. Based on $I_{\text{MINE}}$, the *MINE* estimator $\widehat{I(X, Z)}_n$ is defined as in Eq.2. Estimators for $I_{\text{EB1}}$, $I_{\text{MINE-f}}$ and $I_{\text{EB}}$ can be defined similarly.

$$
\widehat{I(X; Z)}_n = \sup_{\theta \in \Theta} \frac{1}{n} \sum_{i=1}^n T_\theta(x_i, z_i) - \log \frac{1}{n^2} \sum_{i=1}^n \sum_{j=1}^n e^{T_\theta(x_i, z_j)}.
\tag{2}
$$

With infinite samples to approximate expectation, Eq.2 converges to the lower bound $\widehat{I(X, Z)}_\infty = \sup_{\theta \in \Theta} I_{\text{MINE}}$. Note that the number of samples $n$ needs to be substantially more than the number of model parameters $d = |\theta|$ to guarantee that $T_\theta(X, Y)$ does not overfit to the samples $(x_i, z_i)$, $i = 1, 2 \ldots n$ and overestimate MI. Formally, the sample complexity of *MINE* is defined as the minimum number of samples $n$ in order to achieve Eq.3,

$$
\Pr(|\widehat{I(X, Z)}_n - \widehat{I(X, Z)}_\infty| \leq \epsilon) \geq 1 - \delta.
\tag{3}
$$

Specifically, *MINE* proves that under the following assumptions: 1) $T_\theta(X, Z)$ is $L$-Lipschitz; 2) $T_\theta(X, Z) \in [-M, M]$, 3) $\{\theta_i \in [-K, K], \quad \forall i \in 1, \ldots, d\}$, the sample complexity of *MINE* is given by Eq.4.

$$
n \geq \frac{2M^2(d \log(16KL\sqrt{d}/\epsilon) + 2dM + \log(2/\delta))}{\epsilon^2}.
\tag{4}
$$

For example, a neural network with dimension $d = 10,000$, $M = 1$, $K = 0.1$ and $L = 1$, achieving a confidence interval of $\epsilon = 0.1$ with 95% confidence ($\delta = 0.05$) would require $n \geq 18,756,256$ samples. This is achievable for synthetic example generated by GANs like that studied in Belghazi et al. (2018). For real data, however, the cost of data acquisition for reaching statistically significant

---

[1]We follow the same notations in Belghazi et al. (2018). We encourage the review of Belghazi et al. (2018); Poole et al. (2018) for a detailed understanding of $I_{\text{MINE}}$, $I_{\text{EB1}}$, and $I_{\text{EB}}$.

estimation can be prohibitively expensive. Our approach instead uses the MI lower bounds specified in Eq.1 from a prediction perspective, inspired by cross-validation. Our estimator, DEMINE, improves sample complexity by disentangling data for lower bound estimation from data for learning a generalizable $T_\theta(X, Z)$. DEMINE enables high-confidence MI estimation on small datasets.

## 4 APPROACH

Section 4.1 specifies DEMINE for predictive MI estimation and derives the confidence interval; Section 4.2 formulates Meta-DEMINE, explains task augmentation, and defines the optimization algorithms.

### 4.1 PREDICTIVE MUTUAL INFORMATION ESTIMATION

In DEMINE, we interpret the estimation of *MINE-f* lower bound[2] Eq.1 as a learning problem. The goal is given a limited number of samples, infer the optimal network $T_{\theta^*}(X, Z)$ with parameters $\theta^*$ defined as follows:

$$\theta^* = \arg\max_{\theta \in \Theta} \mathbb{E}_{P_{XZ}} T_\theta(X, Z) - \mathbb{E}_{P_X} \mathbb{E}_{P_Z} e^{T_\theta(X,Z)} + 1.$$

Specifically, samples from $P(X, Z)$ are subdivided into a training set $\{(x_i, z_i)_{\text{train}}, i = 1, \ldots, m\}$ and a validation set $\{(x_i, z_i)_{\text{val}}, i = 1, \ldots, n\}$. The training set is used for learning a network $\tilde{\theta}$ as an approximation to $\theta^*$ whereas the validation set is used for computing the DEMINE estimation $\widehat{I(X, Z)}_{n,\tilde{\theta}}$ defined as in Eq.5.

$$\widehat{I(X, Z)}_{n,\tilde{\theta}} = \frac{1}{n} \sum_{i=1}^{n} T_{\tilde{\theta}}(x_i, z_i)_{\text{val}} - \frac{1}{n^2} \sum_{i=1}^{n} \sum_{j=1}^{n} e^{T_{\tilde{\theta}}(x_i, z_j)_{\text{val}}} + 1 \tag{5}$$

We propose an approach to learn $\tilde{\theta}$, DEMINE. DEMINE learns $\tilde{\theta}$ by maximizing the MI lower bound on the training set as follows:

$$\tilde{\theta} = \arg\min_{\theta \in \Theta} \mathcal{L}(\{(x, z)\}_{\text{train}}, \theta), \text{where,}$$

$$\mathcal{L}(\{(x, z)\}_{\mathcal{B}}, \theta) = -\frac{1}{|\mathcal{B}|} \sum_{i=1}^{|\mathcal{B}|} T_\theta(x_i, z_i)_{\mathcal{B}} + \frac{1}{|\mathcal{B}|^2} \sum_{i=1}^{|\mathcal{B}|} \sum_{j=1}^{|\mathcal{B}|} e^{T_\theta(x_i, z_j)_{\mathcal{B}}} - 1. \tag{6}$$

The DEMINE algorithm is shown in Algorithm 2 in appendix.

**Sample complexity analysis.** Because $\tilde{\theta}$ is learned independently of validation samples $\{(x_i, z_i)_{\text{val}}, i = 1, \ldots, n\}$, the sample complexity of the DEMINE estimator does not involve the model class $\mathcal{F}$ and the sample complexity is greatly reduced compared to *MINE-f*. DEMINE estimates $\widehat{I(X, Z)}_{\infty,\tilde{\theta}}$ when infinite number of samples are provided, defined as:

$$\begin{aligned} \widehat{I(X, Z)}_{\infty,\tilde{\theta}} &= \mathbb{E}_{P_{XZ}} T_{\tilde{\theta}}(X, Z) - \mathbb{E}_{P_X} \mathbb{E}_{P_Z} e^{T_{\tilde{\theta}}(X,Z)} + 1 \\ &\leq \sup_{\theta \in \Theta} \mathbb{E}_{P_{XZ}} T_\theta(X, Z) - \mathbb{E}_{P_X} \mathbb{E}_{P_Z} e^{T_\theta(X,Z)} + 1 \leq I(X; Z) \end{aligned} \tag{7}$$

We now derive the sample complexity of DEMINE defined as the number of samples $n$ required for $\widehat{I(X, Z)}_{n,\tilde{\theta}}$ to be a good approximation to $\widehat{I(X, Z)}_{\infty,\tilde{\theta}}$ in Theorem 1.

**Theorem 1.** For $T_{\tilde{\theta}}(X, Z)$ bounded by $[L, U]$, given any accuracy $\epsilon$ and confidence $\delta$, we have:

$$\Pr(|\widehat{I(X, Z)}_{n,\tilde{\theta}} - \widehat{I(X, Z)}_{\infty,\tilde{\theta}}| \leq \epsilon) \geq 1 - \delta$$

when the number of validation samples $n$ satisfies:

$$n \geq n^*, s.t. \ f(n^*) \equiv \min_{0 \leq \xi \leq \epsilon} 2e^{-\frac{2\xi^2 n^*}{(U-L)^2}} + 4e^{-\frac{(\epsilon-\xi)^2 n^*}{2(e^U - e^L)^2}} = \delta \tag{8}$$

---

[2]*MINE* lower bound can also be interpreted in the predictive way, but will result in a higher sample complexity than *MINE-f* lower bound. We choose *MINE-f* in favor of a lower sample complexity over bound tightness.

**Proof.** Since $T_{\tilde{\theta}}(X, Z)$ is bounded by $[L, U]$, applying the Hoeffding inequality to the first half of Eq.5 yields:

$$\Pr(|\frac{1}{n}\sum_{i=1}^{n} T_{\tilde{\theta}}(x_i, z_i) - \mathbb{E}_{P_{XZ}} T_{\tilde{\theta}}(X, Z)| \geq \xi) \leq 2e^{-\frac{2\xi^2 n}{(U-L)^2}}$$

As $e^{T_{\theta}(X,Z)}$ is bounded by $[e^L, e^U]$, applying the Hoeffding inequality twice to the second half of Eq.5:

$$\Pr(|\mathbb{E}_{P_X}\mathbb{E}_{P_Z} e^{T_{\theta}(X,Z)} - \frac{1}{n}\sum_{i=1}^{n}\mathbb{E}_{P_Z}e^{T_{\tilde{\theta}}(x_i, z)}| \geq \zeta) \quad \leq \quad 2e^{-\frac{2\zeta^2 n}{(e^U - e^L)^2}}$$

$$\Pr(|\mathbb{E}_{P_Z}\frac{1}{n}\sum_{i=1}^{n} e^{T_{\theta}(x_i, z)} - \frac{1}{n}\sum_{j=1}^{n}\frac{1}{n}\sum_{i=1}^{n} e^{T_{\tilde{\theta}}(x_i, z_j)}| \geq \zeta) \quad \leq \quad 2e^{-\frac{2\zeta^2 n}{(e^U - e^L)^2}}$$

Combining the above bounds results in:

$$\Pr(|\widehat{I(X, Z)}_{n, \tilde{\theta}} - \widehat{I(X, Z)}_{\infty, \tilde{\theta}}| \leq \xi + 2\zeta) \geq 1 - 2e^{-\frac{2\xi^2 n}{(U-L)^2}} - 4e^{-\frac{2\zeta^2 n}{(e^U - e^L)^2}}$$

By solving $\xi$ to minimize $n$ according to Eq.8 we have:

$$\Pr(|\widehat{I(X, Z)}_{n, \tilde{\theta}} - \widehat{I(X, Z)}_{\infty, \tilde{\theta}}| \leq \epsilon) \geq 1 - \delta. \qquad \blacksquare$$

Theorem 1 also implies the following MI lower confidence interval under limited number of samples

$$\Pr(I(X; Z) \geq \widehat{I(X, Z)}_{n, \tilde{\theta}} - \epsilon) \geq 1 - \delta$$

Compared to *MINE*, as per the example shown in Section 3, for $M = 1$ (*i.e.* $L = -1$ and $U = 1$), $\delta = 0.05$, $\epsilon = 0.1$, our estimator requires $n = 10,742$ compared to *MINE* requiring $n = 18,756,256$ *i.i.d* validation samples to estimate a lower bound, which makes MI-based dependency analysis feasible for domains where data collection is prohibitively expensive, *e.g.* fMRI scans. In practice, sample complexity can be further optimized by optimizing hyperparameters $U$ and $L$.

Note that unlike Eq.3, Theorem 1 bounds the closeness of the DEMINE estimate, $\widehat{I(X, Z)}_{n, \tilde{\theta}}$, not towards the MI lower bound $\sup_{\theta \in \Theta} I_{\text{MINE-f}}$, but towards the MI lower bound $\widehat{I(X, Z)}_{\infty, \tilde{\theta}}$. Therefore, the sample complexity of DEMINE as in Eq.8 makes fair comparison with the sample complexity of *MINE* as in Eq.4. *MINE*'s higher sample complexity stems from the need to bound the generalization error of $T_{\theta}(X, Z)$ on unseen $\{(x, z)\}$. Existing generalization bounds are known to be overly loose, as over-parameterized neural networks have been shown to generalize well in classification and regression tasks (Zhang et al., 2016). By using a learning-based formulation, DEMINE not only avoids the need to bound generalization error, but also allows further generalization improvements by learning $\tilde{\theta}$ through meta-learning.

In the following section, we present a meta-learning formulation, Meta-DEMINE, that learns $\tilde{\theta}$ for generalization given the same model class and training samples.

## 4.2 META-LEARNING

Given training data $\{(x_i, z_i)_{\text{train}}, i = 1, \ldots m\}$, Meta-DEMINE first generates MI estimation tasks each consisting of a meta-training split A and a meta-val split B through a novel *task augmentation* process. And then a parameter initialization $\theta_{\text{init}}$ is then learned to maximize MI estimation performance on the generated tasks using initialization $\theta_{\text{init}}$ as shown in Eq.9.

$$\theta_{\text{init}} = \arg\min_{\theta^{(0)} \in \Theta} \mathbb{E}_{(A,B) \in \mathcal{T}} \mathcal{L}((x, z)_B, \theta^{(t)}), \text{with}, \theta^{(t)} \equiv \text{MetaTrain}((x, z)_A, \theta^{(0)}). \tag{9}$$

Here $\theta^{(t)} = \text{MetaTrain}((x, z)_A, \theta^{(0)})$ is the meta-training process of starting from an initialization $\theta^{(0)}$ and applying Stochastic Gradient Descent (SGD) [3] over $t$ steps to learn $\theta$ where in every meta training iteration we have:

$$\theta^{(t)} \leftarrow \theta^{(t-1)} - \gamma \nabla \mathcal{L}((x, z)_A, \theta^{(t-1)}).$$

---

[3]In practice, the Adam optimizer (Kingma & Ba, 2014) is used for faster optimization. The Adam optimizer uses first and second order momentums of the gradient to speed up optimization. Illustrating SGD for simplicity.

Finally, $\tilde{\theta}$ is learned using the entire training set $\{(x_i, z_i)_{\text{train}}, i = 1, \ldots, m\}$ with $\theta_{\text{init}}$ as initialization:

$$\tilde{\theta} = \text{MetaTrain}\big((x, z)_{\text{train}}, \theta_{\text{init}}\big).$$

**Task Augmentation:** Meta-DEMINE adapts MAML (Finn et al., 2017a) for MI lower bound maximization. MAML has been shown to improve generalization performance in $N$-class $K$-shot image classification. MI estimation, however, does not come with predefined classes and tasks. A naive approach to produce tasks would be through cross-validation – partitioning training data into meta-training and meta-validation splits. However, merely using cross-validation tasks is prone to overfitting – a $\theta_{\text{init}}$, which memorizes all training samples would as a result have memorized all meta-validation splits. Instead, Meta-DEMINE generates tasks by augmenting the cross-validation tasks through *task augmentation*. Training samples are first split into meta-training and meta-validation splits, and then transformed using the same random invertible transformation to increase task diversity. Meta-DEMINE generates invertible transformation by sequentially composing the following functions:

$$
\begin{array}{lll}
\textit{Mirror}: & m(x) = (2n - 1)x, & n \sim \text{Bernoulli}(\tfrac{1}{2}), \\
\textit{Permute}: & P(x) = {}^n P_d, & \text{Permute dimensions.} \\
\textit{Offset}: & O(x) = x + \epsilon, & \epsilon \sim \mathcal{U}(-0.1, 0.1), \\
\textit{Gamma}: & G(x) = \text{sign}(x)\,|x|^{\gamma}, & \gamma \sim \mathcal{U}(0.5, 2),
\end{array}
$$

Since the MI between two random variables is invariant to invertible transformations on each variable, $\text{MetaTrain}(\cdot, \cdot)$ is expected to arrive at the same MI lower bound estimation regardless of the transformation applied. At the same time, memorization is greatly suppressed, as the same pair $(x, z)$ can have different $\log \frac{p(x,z)}{p(x)p(z)}$ under different transformations. More sophisticated invertible transformations (affine, piece-wise linear) can also be added. Task augmentation is an orthogonal approach to data augmentation. Using image classification as an example, data augmentation generates variations of the image, translated, or rotated images assuming that they are valid examples of the class. Task augmentation on the other hand, does not make such an assumption. Task augmentation requires the initial parameters $\theta_{\text{init}}$ to be capable of recognizing the same class in a world where all images are translated and/or rotated, with the assumption that the optimal initialization should easily adapt to both the upright world and the translated and/or rotated world.

**Optimization:** Solving $\theta_{\text{init}}$ using the meta-learning formulation Eq.9 poses a challenging optimization problem. The commonly used approach is back propagation through time (BPTT) which computes second order gradients and directly back propagates gradients from $\text{MetaTrain}((x, z)_{\text{A}}, \theta^{(0)})$ to $\theta_{\text{init}}$. BPTT is very effective for a small number of optimization steps, but is vulnerable to exploding gradients and is memory intensive. In addition to BPTT, we find that stochastic finite difference algorithms such as Evolution Strategies (ES) (Salimans et al., 2017) and Parameter-Exploring Policy Gradients (PEPG) (Sehnke et al., 2010) can sometimes improve optimization robustness. In practice, we switch betwen BPTT and PEPG depending on the number of meta-training iterations. Meta-DEMINE algorithm is specified in Algorithm 1.

## 5 EVALUATION ON SYNTHETIC DATASETS

**Dataset.** We evaluate our approaches DEMINE and Meta-DEMINE against baselines and state-of-the-art approaches on 3 synthetic datasets: 1D Gaussian, 20D Gaussian and sine wave. For 1D and 20D Gaussian datasets, following Belghazi et al. (2018), we define two $k$-dimensional multivariate Gaussian random variables $X$ and $Z$ which have component-wise correlation $corr(X_i, Z_j) = \delta_{ij}\rho$, where $\rho \in (-1, 1)$ and $\delta_{ij}$ is Kronecker's delta. Mutual information $I(X; Z)$ has a closed form solution $I(X; Z) = -k \ln(1 - \rho^2)$. For the sine wave dataset, we define two random variables $X$ and $Z$, where $X \sim \mathcal{U}(-1, 1)$, $Z = \sin(aX + \frac{\pi}{2}) + 0.05\epsilon$, and $\epsilon \sim \mathcal{N}(0, 1)$. Estimating mutual information accurately given few pairs of $(X, Z)$ requires the ability to extrapolate the sine wave given few examples. Ground truth MI for sine wave dataset is approximated by running the the KSG Estimator (Kraskov et al., 2004) on $1,000,000$ samples.

**Implementation.** We compare our estimators, DEMINE and Meta-DEMINE, against the KSG estimator (Kraskov et al., 2004) MI-KSG and MINE-f (Belghazi et al., 2018). For both DEMINE and Meta-DEMINE, we study variance reduction mode, referred to as *-vr*, where hyperparameters are selected by optimizing 95% confident estimation mean ($\mu - 2\sigma_\mu$) and statistical significance mode, referred to as *-sig*, where hyperparameters are selected by optimizing 95% confident MI

---

**Algorithm 1** Meta-DEMINE

**Input Data**: $\{(x,z)_\text{train}, (x,z)_\text{val}\}$
**Parameters**: batch $\mathcal{B}$, Meta Learning Iterations $N_M$, Task Augmentation Iterations $N_T$, Optimization Iterations $N_O$, Ratio $r$, Learning rate $\eta$, Meta Learning Rate $\eta_\text{meta}$
**Output**: MI, $T_{\theta_\text{init}}(X,Z), T_\theta(X,Z)$

1: **for** i = 1 : $N_M$ **do**
2:     **for** j = 1 : $N_T$ **do**
3:         $A = r \times \text{train}, B = \text{train} - A$
4:         Split $(x,z)_\text{train}$ into $(x,z)_\text{A}$ and $(x,z)_\text{B}$
5:         Transformation $R_x$ for $x$, $R_x(\cdot) = \text{m}(\text{P}(\text{O}(\text{G}(\cdot))))$
6:         Transformation $R_z$ for $z$, $R_z(\cdot) = \text{m}(\text{P}(\text{O}(\text{G}(\cdot))))$
7:         $\theta_\text{meta}^{(0)} \leftarrow \theta_\text{init}$
8:         **for** k = 1 : $N_O$ **do**
9:             Sample a batch of $(x,z)_\mathcal{B} \sim (x,z)_\text{A}$
10:            Compute $\mathcal{L}\big((R_x(x), R_z(z))_\mathcal{B}, \theta_\text{meta}^{(k)}\big)$
11:            Compute $\nabla_{\theta_\text{meta}^{(k)}}\mathcal{L}$ – gradient for $\theta_\text{meta}$
12:            Update $\theta_\text{meta}$ using Adam Kingma & Ba (2014) with $\eta$
13:         **end for**
14:         Compute $\mathcal{L}_\text{meta}\big((R_x(x), R_z(z))_\text{B}, \theta_\text{meta}^{(N_O)}\big)$
15:         Compute $\nabla_{\theta_0}\mathcal{L}_\text{meta}$ – gradient to $\theta_\text{init}$ using BPTT
16:     **end for**
17:     Update $\theta_\text{init}$ using Adam Kingma & Ba (2014) with $\eta_{meta}$
18: **end for**
19: $\theta^{(0)} \leftarrow \theta_\text{init}$
20: **for** i = 1 : $N_O$ **do**
21:     Sample a batch of $(x,z)_\mathcal{B} \sim (x,z)_\text{train}$
22:     Compute $\mathcal{L}\big((x,z)_\mathcal{B}, \theta^{(i)}\big)$
23:     Compute gradient $\nabla_\theta\mathcal{L}$
24:     Update $\theta$ using Adam with $\eta$
25: **end for**
26: Compute MI $= \mathcal{L}\big((x,z)_\text{val}, \theta^{(N_O)}\big)$
27: **return** MI, $\theta_\text{init}, \theta^{(N_O)}$

---

lower bound $(\mu - \epsilon)$. Samples $(x,z)$ are split 50%-50% into $(x,z)_\text{train}$ and $(x,z)_\text{val}$. We use a separable network architecture $T_\theta(x,z) = M\big(\tanh(w\cos\langle f(x), g(z)\rangle + b) - t\big)$. $f$ and $g$ are MLP encoders that embed signals $x$ and $z$ into vector embeddings. Hyperparameters $t \in [-1,1]$ and $M$ control upper and lower bounds $T_\theta(x,z) \in [-M(1+t), M(1-t)]$. Parameters $w$ and $b$ are learnable parameters. MLP design and optimization hyperparameters are selected using Bayesian hyperparameter optimization (Bergstra et al., 2013) described below.

Hyperparameter search on DEMINE-vr and DEMINE-sig was conducted using the hyperopt package [4]. Seven hyperparameters were involved in hyperparameter search: 1) number of encoder layers $[1, 5]$, 2) encoder hidden size $[8, 256]$, 3) learning rate $\eta$ $[10^{-4}, 3 \times 10^{-1}]$ in log scale, 4) number of optimization iterations $N_O$ $[5, 200]$ (sine wave $[5, 5000]$) in log scale, 5) batch size $\mathcal{B}$ $[256, 1024]$, 6) $M$, $[10^{-3}, 5]$ in log scale, 7) $t$, $[-1, 1]$. Mean $\mu$ and sample standard deviation $\sigma$ of MI estiamte computed over 3-fold cross-validation on $(x,z)_\text{train}$. DEMINE-vr maximizes two sigma low $\mu - 2\sigma_\mu$ where $\sigma_\mu = \frac{1}{\sqrt{3}}\sigma$ due to 3-fold cross-validation. DEMINE-sig maximizes statistical significance $\mu - \epsilon$ where $\epsilon$ is two-sided 95% confidence interval of MI. Meta-DEMINE-vr and Meta-DEMINE-sig subsequently reuse these hyperparameters as DEMINE-vr and DEMINE-sig.

Meta-learning hyperparameters are chosen as outer loop $N_M = 3,000$ iterations, task augmentation $N_T = 1$ iterations, $r = 0.8$, $\eta_\text{meta} = \frac{\eta}{3}$, with task augmentation mode $m(P(O(\cdot)))$. $N_O$ was capped at 30 iterations for 1D and 20D Gaussian datasets due to memory limit. For the sine wave datasets with large $N_O$, we used PEPG (Sehnke et al., 2010) rather than BPTT.

---

[4]Hyperopt package: `https://github.com/hyperopt/hyperopt`.

For MI-KSG, we use off-the-shelf implementation by Gao et al. (2017) with default number of nearest neighbors $k = 3$. MI-KSG does not provide any confidence interval. For MINE-f, we use the same network architecture same as DEMINE-vr. we implement both the original formulation which optimizes $T_\theta$ on $(x, z)$ till convergence (10k iters), as well as our own implementation MINE-f-ES with early stopping, where optimization is stopped after the same number of iterations as DEMINE-vr to control overfitting.

**Results.** Figure 1(a) shows MI estimation performance on 20D Gaussian datasets with varying $\rho \in \{0, 0.1, 0.2, 0.3, 0.4, 0.5\}$ using $N = 300$ samples. Results are averaged over 5 runs to compare estimator bias, variance and confidence. Note that Meta-DEMINE-sig detects the highest $p < 0.05$ confidence MI, outperforming DEMINE-sig which is a close second. Both detect $p < 0.05$ statistically significant dependency starting $\rho = 0.3$, whereas estimations of all other approaches are low confidence. It shows that in contrary to common belief, estimating the variational lower bounds with high confidence can be challenging under limited data. MINE-f estimates MI $> 3.0$ and MINE-f-ES estimates positive MI when $\rho = 0$, both due to overfitting, despite MINE-f-ES having the lowest empirical bias. DEMINE variants have relatively high empirical bias but low variance due to tight upper and lower bound control, which provides a different angle to understand bias-variance trade off in MI estimation (Poole et al., 2018).

Figure 1(b,c,d) shows MI estimation performance on 1D, 20D Gaussian and sine wave datasets with fixed $\rho = 0.8, 0.3$ and $a = 8\pi$ respectively, with varying $N \in \{30, 100, 300, 1000, 3000\}$ number of samples. More samples asymptotically improves empirical bias across all estimators. As opposed to 1D Gaussian datasets which are well solved by $N = 300$ samples, higher-dimensional 20D Gaussian and higher-complexity sine wave datasets are much more challenging and are not solved using $N = 3000$ samples with a signal-agnostic MLP architecture. DEMINE-sig and Meta-DEMINE-sig detect $p < 0.05$ statistically significant dependency on not only 1D and 20D Gaussian datasets where $x$ and $z$ have non-zero correlation, but also on the sine wave datasets where correlation between $x$ and $z$ is 0. This means that DEMINE-sig and Meta-DEMINE-sig can be used for nonlinear dependency testing to complement linear correlation testing.

We study the effect of cross-validation meta-learning and task augmentation on 20D Gaussian with $\rho = 0.3$ and $N = 300$. Figure 2 plots performance of Meta-DEMINE-vr over $N_M = 3000$ meta iterations under combinations of task augmentations modes and number of adaptation iterations $N_O \in \{0, 20\}$. Overall, task augmentation modes which involve axis flipping $m(\cdot)$ and permutation $P(\cdot)$ are the most successful. With $N_O = 20$ steps of adaptation, task augmentation modes $P(\cdot)$, $m(P(\cdot))$ and $m(P(O(\cdot)))$ prevent overfitting and improves performance. The performance improvements of task augmentation is not simply from change in batch size, learning rate or number of optimization iterations, because meta-learning without task augmentation for both $N_O = 0$ and 20 could not outperform baseline. Meta-learning without task augmentation and with task augmentation but using only $O(\cdot)$ or $G(\cdot)$ result in overfitting. Task augmentation with $m(\cdot)$ or $m(P(O(G(\cdot))))$ prevent overfitting, but do not provide performance benefits, possibly because their complexity is insufficient or excessive for 20 adaptation steps. Further more, task augmentation with no adaptation ($N_O = 0$) falls back to data augmentation, where samples from transformed distributions are directly used to learn $T_\theta(x, z)$. Data augmentation with $O(\cdot)$ outperforms no augmentation, but is unable to outperform baseline and suffers from overfitting. It shows that task augmentation provides improvements orthogonal to data augmentation.

# 6 APPLICATION: FMRI INTER-SUBJECT CORRELATION (ISC) ANALYSIS

Humans use language to effectively transmit brain representations among conspecifics. For example, after witnessing an event in the world, a speaker may use verbal communication to evoke neural representations reflecting that event in a listener's brain (Hasson et al., 2012). The efficacy of this transmission, in terms of listener comprehension, is predicted by speaker-listener neural synchrony and synchrony among listeners (Stephens et al., 2010). To date, most work has measured brain-to-brain synchrony by locating statistically significant inter-subject correlation (ISC); quantified as the Pearson product-moment correlation coefficient between response time series for corresponding voxels or regions of interest (ROIs) across individuals (Hasson et al., 2004; Schippers et al., 2010; Silbert et al., 2014; Nastase et al., 2019). Using DEMINE and Meta-DEMINE for statistical dependency testing, we can extend ISC analysis to capture nonlinear and higher-order interactions in continuous fMRI responses. Specifically, given synchronized fMRI response frames in two brain

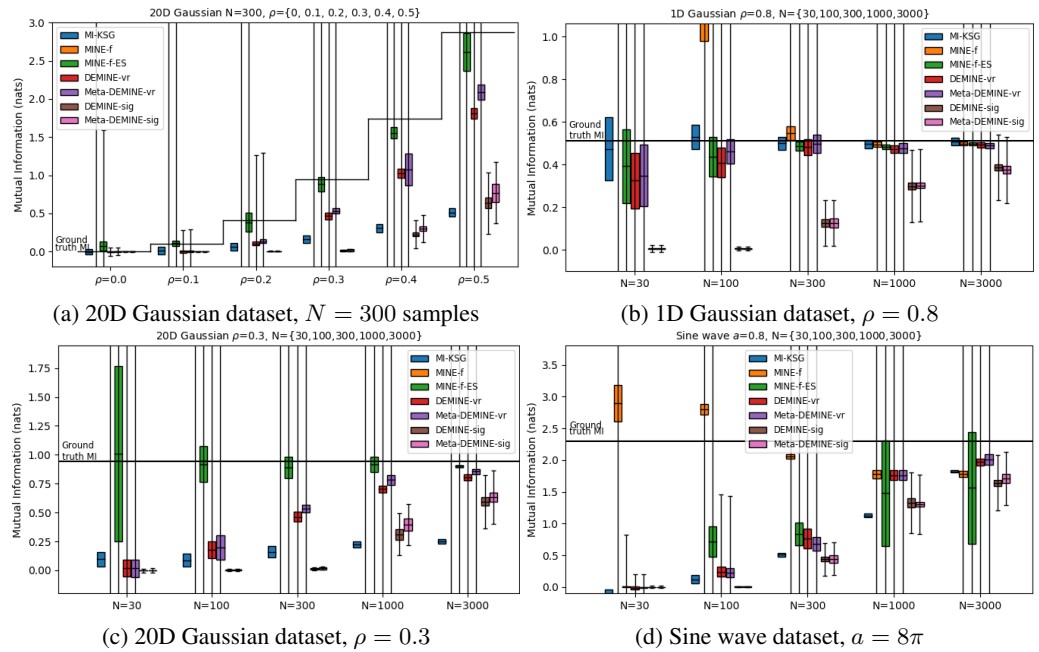

(a) 20D Gaussian dataset, $N = 300$ samples

(b) 1D Gaussian dataset, $\rho = 0.8$

(c) 20D Gaussian dataset, $\rho = 0.3$

(d) Sine wave dataset, $a = 8\pi$

Figure 1: Comparing MI Estimation performance of DEMINE and Meta-DEMINE with the KSG estimator Kraskov et al. (2004) and MINE-f Belghazi et al. (2018) on different datasets using varying number of samples. The bars show estimator mean and standard deviation averaged over 5 runs with different seeds. The error bars show 95% confidence interval (not available for MI-KSG). The statistical significance focused variants DEMINE-sig and Meta-DEMINE-sig achieves the highest 95% confident MI estimation. Meta-DEMINE improves over DEMINE most of the time.

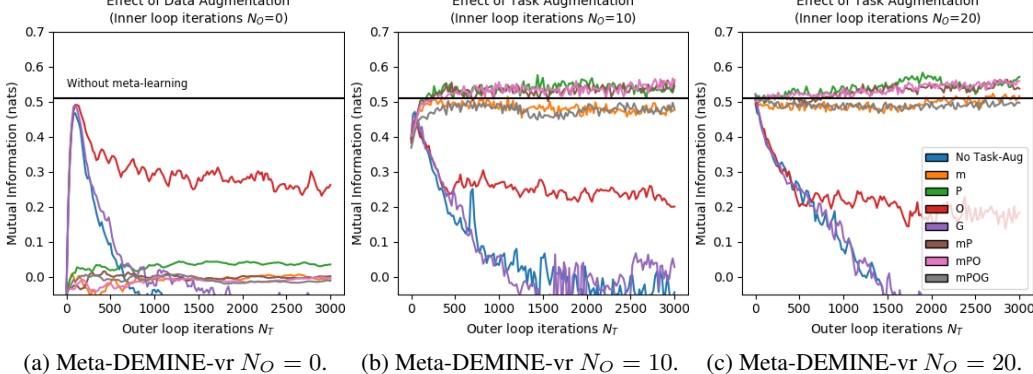

(a) Meta-DEMINE-vr $N_O = 0$.  (b) Meta-DEMINE-vr $N_O = 10$.  (c) Meta-DEMINE-vr $N_O = 20$.

Figure 2: To study the effect of task augmentation and number of adaptation steps, we run Meta-DEMINE-vr with different task augmentation modes and vary number of adaptation iterations $N_O \in \{0, 10, 20\}$ on Gaussian 20D, $\rho = 0.3$ dataset. Combinations of permutation and mirroring operations are effective in reducing overfitting and improving performance.

Table 1: Number of HCP-MMP1 regions with significant correlation (r) and MI (DEMINE, Meta-DEMINE) during listening.

| No. shared | r | DEMINE -sig | Meta -DEMINE -sig |
|---|---|---|---|
| r | 37 | 24 | 23 |
| DEMINE-sig | 24 | 28 | 26 |
| Meta-DEMINE-sig | 23 | 26 | 29 |

Table 2: Segment classification accuracy for NeuralMI versus Pearson's correlation in 1-vs-rest*.

| Classification Accuracy (%) | ISC Mask | | | | | dDMN Mask | | | | |
|---|---|---|---|---|---|---|---|---|---|---|
| | P | F | Br | Bk | MI | P | F | Br | Bk | MI |
| Chance | 3.7 | 1.8 | 2.6 | 1.9 | N/A | 3.7 | 1.8 | 2.6 | 1.9 | N/A |
| Pearson's r 1vR | 35.0 | 20.4 | 25.8 | 31.5 | N/A | 14.8 | 6.4 | **11.8** | 9.9 | N/A |
| DEMINE-vr 1vR | 42.8 | 28.0 | 32.8 | 35.9 | 0.637 | **16.5** | **7.9** | 11.6 | **12.0** | **0.035** |
| Meta-DEMINE-vr 1vR | **47.2** | **32.5** | **39.9** | **41.0** | **0.752** | 13.7 | 7.9 | 8.2 | 8.9 | 0.031 |

Abbreviations: P: Pieman; F: Forgot; Br: Bronx; Bk: Black, MI: Mutual Information.
*Note that all the results are averaging over the subjects.

regions $X$ and $Z$ across $K$ subjects $X_i, Z_i, i = 1, \ldots, K$ as random variables. We model the conditional mutual information $I(X_i; Z_j | i \neq j)$ as the MI form of pair-wise ISC analysis. By definition, $I(X_i; Z_j | i \neq j)$ first computes MI between activations $X_i$ and $Z_j$ from subjects $i$ and $j$ respectively, and then average across pairs of subjects $i \neq j$. It can be lower bounded using Eq. 7 by learning a $T_\theta(x, z)$ shared across all subject pairs.

**Dataset.** We study MI-based and correlation-based ISC on a fMRI story comprehension dataset by Nastase et al. (2019) with 40 participants listening to four spoken stories. Average story duration is 11 minutes. An fMRI frame with full brain coverage is captured at repetition time 1 TR =1.5 seconds with 2.5mm isotropic spatial resolution. We restricted our analysis to subsets of voxels defined using independent data from previous studies: functionally-defined masks of high ISC voxels (ISC; 3,800 voxels) and dorsal Default-Mode Network voxels (dDMN; 3,940 voxels) from Simony et al. (2016) as well as 180 HCP-MMP1 multimodal cortex parcels from Glasser et al. (2016). All masks were defined in MNI space.

**Implementation.** We compare MI-based ISC using DEMINE and Meta-DEMINE with correlation-based ISC using Pearson's correlation. DEMINE and Meta-DEMINE setup follows Section 5. The fMRI data were partitioned by subject into a train set of 20 subjects and a validation set of 20 different subjects. Residual 1D CNN is used instead of MLP as the encoder for studying temporal dependency. For Pearson's correlation, high-dimensional signals are reshaped to 1D for correlation analysis. Effective sample size for confidence interval calculation is the number of unique non-overlapping fMRI samples.

**Results.** We first examine, for the fine grained HCM-MMP1 brain regions, which have $p < 0.05$ statistically significant MI and Pearson's correlation. Table 1 shows the result. Overall, more regions have statistically significant correlation than dependency. This is expected because correlation requires less data to detect. But Meta-DEMINE is able to find 6 brain regions that have statistically significant dependency but lacks significant correlation. This shows that MI analysis can be used to complement correlation-based ISC analysis.

By considering temporal ISC over time, fMRI signals can be modeled with improved accuracy. In Table 2 we apply DEMINE and Meta-DEMINE with $L = 10$TRs (15s) sliding windows as random variables to study amount of information that can be extracted from ISC and dDMN masks. We use between-subject time-segment classification (BSC) for evaluation (Haxby et al., 2011; Guntupalli et al., 2016). Each fMRI scan is divided into $K$ non-overlapping $L = 10$ TRs time segments. The BSC task is one versus rest retrieval: retrieve the corresponding time segment $z$ of an individual given a group of time segments $x$ excluding that individual, measured by top-1 accuracy. For retrieval score, $T_\theta(X, Z)$ is used for DEMINE and Meta-DEMINE and $\rho(X, Z)$ is used for Pearson's correlation as a simple baseline. With CNN as encoder, DEMINE and Meta-DEMINE model the signal better and achieve higher accuracy. Also. Meta-DEMINE is able to extract 0.75 nats of MI from the ISC mask over 10 TRs or 15s, which could potentially be improved by more samples.

## 7 CONCLUSION

We illustrated that a predictive view of the MI lower bounds coupled with meta-learning results in data-efficient variational MI estimators, DEMINE and Meta-DEMINE, that are capable of performing statistical test of dependency. We also showed that our proposed task augmentation reduces overfitting and improves generalization in meta-learning. We successfully applied MI estimation to real world, data-scarce, fMRI datasets. Our results suggest a greater avenue of using neural networks and meta-learning to improve MI analysis and applying neural network-based information theory tools to enhance the analysis of information processing in the brain. Model-agnostic, high-confidence, MI lower bound estimation approaches – including *MINE*, DEMINE and Meta-DEMINE– are limited to estimating small MI lower bounds up to $O(\log N)$ as pointed out in (McAllester & Statos, 2018), where $N$ is the number of samples. In real fMRI datasets, however, strong dependency is rare and existing MI estimation tools are limited more by their ability to accurately characterize the dependency. Nevertheless, when quantitatively measuring strong dependency, cross-entropy (McAllester & Statos, 2018) or model-based quantities, alternatives to MI, such as correlation or CCA, may be measured with high confidence.

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

# A APPENDIX

## A.1 THE DEMINE ALGORITHM

---

**Algorithm 2** DEMINE

---

    **Input Data**: $\{(x, z)_\text{train}, (x, z)_\text{val}\}$
    **Parameters**: Batch $\mathcal{B}$, Iterations $N_O$, Learning rate $\eta$
    **Output**: MI, $T_\theta(X, Z)$
1:  $\theta^{(0)} \leftarrow$ Xavier Initialization (Glorot & Bengio, 2010)
2: **for** i = 1 : $N_O$ **do**
3:     Sample a batch of $(x_i, z_i)_\mathcal{B} \sim (x, z)_\text{train}$
4:     Compute $\mathcal{L}\left((x_i, z_i)_\mathcal{B}, \theta^{(i-1)}\right)$
5:     Compute $\nabla_\theta^{(i)} \mathcal{L}$ – gradient for $\theta$
6:     Update $\theta^{(i)}$ using Adam (Kingma & Ba, 2014) with $\eta$
7: **end for**
8: MI $= \widehat{I(X, Z)}_{n, \theta^{(N_O)}}$
9: **return** MI, $\theta^{(N_O)}$

---

## A.2 ADDITIONAL DETAILS OF THE FMRI DATASET

The dataset we used contains 40 participants (mean age = 23.3 years, standard deviation = 8.9, range: 1853; 27 female) recruited to listen to four spoken stories[5][6]. The stories were renditions of "Pie Man" and "Running from the Bronx" by Jim OGrady (O'Grady, 2018b;a), "The Man Who Forgot Ray Bradbury" by Neil Gaiman (Gaiman, 2018), and "I Knew You Were Black" by Carol Daniel (Daniel, 2018); story durations were 7, 9, 14, and 13 minutes, respectively. After scanning, participants completed a questionnaire comprising 25-30 questions per story intended to measure narrative comprehension. The questionnaires included multiple choice, True/False, and fill-in-the-blank questions, as well as four additional subjective ratings per story. Functional and structural images were acquired using a 3T Siemens Prisma with a 64-channel head coil. Briefly, functional images were acquired in an interleaved fashion using gradient-echo echo-planar imaging with a multiband acceleration factor of 3 (TR/TE = 1500/31 ms where TE stands for "echo time", resolution = 2.5 mm isotropic voxels, full brain coverage).

All fMRI data were formatted according to the Brain Imaging Data Structure (BIDS) standard (Gorgolewski et al., 2016) and preprocessed using the fMRIPrep library (Esteban et al., 2018). Functional data were corrected for slice timing, head motion, and susceptibility distortion, and normalized to MNI space using nonlinear registration. Nuisance variables comprising head motion parameters, framewise displacement, linear and quadratic trends, sine/cosine bases for high-pass filtering (0.007 Hz), and six principal component time series from cerebrospinal fluid (CSF) and white matter (WM) were regressed out of the signal using the Analysis of Functional NeuroImages (AFNI) software suite (Cox, 1996).

The fMRI data comprise $\mathcal{X} \in \mathbb{R}^{V_i \times T}$ for each subject, where $V_i$ represents the flattened and masked voxel space and $T$ represents the number of samples (in TRs) during auditory stimulus presentation.

**Additional Details on Dataset Collection** Functional and structural images were acquired using a 3T Siemens Magnetom Prisma with a 64-channel head coil. Functional, blood-oxygenation-level-dependent (BOLD) images were acquired in an interleaved fashion using gradient-echo echo-planar imaging with pre-scan normalization, fat suppression, a multiband acceleration factor of 3, and no in-plane acceleration: TR/TE = 1500/31 ms, flip angle = 67°, bandwidth = 2480 hz per pixel, resolution = 2.5 $mm^3$ isotropic voxels, matrix size = 96 x 96, Field of view (FoV) = 240 x 240 mm, 48 axial slices with roughly full brain coverage and no gap, anteriorposterior phase encoding. At the beginning of each scanning session, a T1-weighted structural scan (where T1 stands for

---

[5]Two of the stories were told by a professional storyteller undergoing an fMRI scan; however, fMRI data for the speaker were not analyzed for the present work due to the head motion induced by speech production.

[6]The study was conducted in compliance with the Institutional Review Board of the University

"longitudinal relaxation time"), was acquired using a high-resolution single-shot Magnetization-Prepared 180 degrees radio-frequency pulses and RApid Gradient-Echo (MPRAGE) sequence with an in-plane acceleration factor of 2 using GeneRalized Autocalibrating Partial Parallel Acquisition (GRAPPA): TR/TE/TI = 2530/3.3/1100 ms where TI stands for inversion time, flip angle = 7°, resolution = 1.0 x 1.0 x 1.0 mm voxels, matrix size = 256 x 256, FoV = 256 x 256 x 176 mm, 176 sagittal slices, ascending acquisition, anteriorposterior phase encoding, no fat suppression, 5 min 53 s total acquisition time. At the end of each scanning session a T2-weighted (where T2 stands for "transverse relaxation time") structural scan was acquired using the same acquisition parameters and geometry as the T1-weighted structural image: TR/TE = 3200/428 ms, 4 minutes 40 seconds total acquisition time. A field map was acquired at the beginning of each scanning session, but was not used in subsequent analyses.

**Additional Details on Dataset Preprocessing** Preprocessing was performed using the fMRIPrep library[7] Esteban et al. (2018), a Nipype library[8] (Gorgolewski et al., 2011) based tool. T1-weighted images were corrected for intensity non-uniformity using the N4 bias field correction algorithm (Tustison et al., 2010) and skull-stripped using Advanced Normalization Tools (ANTs) (Avants et al., 2008). Nonlinear spatial normalization to the International Consortium for Brain Mapping (ICBM) 152 Nonlinear Asymmetrical template version 2009c (Fonov et al., 2009) was performed using ANTs. Brain tissue segmentation cerebrospinal fluid, white matter, and gray matter was was performed using FSL library's[9] FAST tool Zhang et al. (2001). Functional images were slice timing corrected using AFNI software's 3dTshift (Cox, 1996) and corrected for head motion using FSL library's MCFLIRT tool (Jenkinson et al., 2002). "Fieldmap-less" distortion correction was performed by co-registering each subject's functional image to that subject's intensity-inverted T1-weighted image (Wang et al., 2017) constrained with an average field map template (Treiber et al., 2016). This was followed by co-registration to the corresponding T1-weighted image using FreeSurfer software's[10] boundary-based registration (Greve & Fischl, 2009) with 9 degrees of freedom. Motion correcting transformations, field distortion correcting warp, BOLD-to-T1 transformation and T1-to-template (MNI) warp were concatenated and applied in a single step with Lanczos interpolation using ANTs. Physiological noise regressors were extracted applying "a Component Based Noise Correction Method" aCompCor (Behzadi et al., 2007). Six principal component time series were calculated within the intersection of the subcortical mask and the union of CSF and WM masks calculated in T1w (T1 weighted) space, after their projection to the native space of each functional run. Framewise displacement (Power et al., 2014) was calculated for each functional run. Functional images were downsampled to 3 mm resolution. Nuisance variables comprising six head motion parameters (and their derivatives), framewise displacement, linear and quadratic trends, sine/cosine bases for high-pass filtering (0.007 Hz cutoff), and six principal component time series from an anatomically-defined mask of cerebrospinal fluid and white matter were regressed out of the signal using AFNI's 3dTproject (Cox, 1996). Functional response time series were z-scored for each voxel.

### A.3   COMPARISON BETWEEN HSIC AND DEMINE

We first review the Hilbert-Schmidt independence criterion (HSIC), a widely-studied correlation-based independence criterion and discuss its connections with the MINE family of mutual information lower bound methods, and then study DEMINE and a spectral HSIC implementation on the synthetic datasets.

The HSIC approach (Gretton et al., 2005b;a) is based on a necessary and sufficient condition of independence: two random variables $X$ and $Z$ are independent if and only if for all bounded or positive functions $f$ and $g$ $\mathbb{R}^d \mapsto \mathbb{R}$, $\mathbb{E}_{XZ}f(X)g(Z) - \mathbb{E}_X f(X)\mathbb{E}_Z g(Z) = 0$, or equivalently $\mathbb{E}_{XZ}(f(X) - E_X f(X))(g(Z) - E_Z g(Z)) = 0$. A proof can be constructed by showing equivalence to the definition of independence, $P(X, Z) = P(X)P(Z)$.

To construct an independence test, existing approaches (Gretton et al., 2005b;a) use Reproducing Kernel Hilbert Spaces (RKHS) for $f$ and $g$, a function space that not only covers all functions between $[0, 1]$, but also allows computationally efficient estimation or bounding of

---

[7] https://github.com/poldracklab/fmriprep
[8] https://github.com/nipy/nipype
[9] https://fsl.fmrib.ox.ac.uk/fsl/fslwiki/FSL
[10] https://surfer.nmr.mgh.harvard.edu/fswiki/FreeSurferWiki

$\text{COCO}(X, Z) = \sup_{f,g} \mathbb{E}_{XZ} f(X)g(Z) - \mathbb{E}_X f(X)\mathbb{E}_Z g(Z)$

given samples, and test $\text{COCO}(X, Z) = 0$. Confidence intervals are derived through McDiarmid's inequality, or using closed-form distributions to approximate the test statistics to a certain order of moments, and compute the confidence interval from the closed-form distribution.

The $\text{COCO}(X, Z)$ used by HSIC estimators bears great resemblance to the MINE family of mutual information estimators. In fact, it can be shown that

$$
\begin{aligned}
\text{COCO}(X, Z) &= \sup_{f,g} \mathbb{E}_{(x,z)\sim P_{XZ}} f(x)g(z) - \mathbb{E}_{x\sim P_X} f(x)\mathbb{E}_{Z\sim P_Z} g(z) \\
&= \sup_{f,g} \mathbb{E}_{(x,z)\sim P_{XZ}} f(x)g(z) - \mathbb{E}_{x\sim P_X, z\sim P_Z} \log e^{f(x)g(z)} \\
&\geq \sup_{f,g} \mathbb{E}_{(x,z)\sim P_{XZ}} f(x)g(z) - \mathbb{E}_{x\sim P_X} \log \mathbb{E}_{z\sim P_Z} e^{f(x)g(z)} \approx I_{\text{EB1}} \\
&\geq \sup_{f,g} \mathbb{E}_{(x,z)\sim P_{XZ}} f(x)g(z) - \log \mathbb{E}_{x\sim P_X, z\sim P_Z} e^{f(x)g(z)} \approx I_{\text{MINE}} \\
&\geq \sup_{f,g} \mathbb{E}_{(x,z)\sim P_{XZ}} f(x)g(z) - \mathbb{E}_{x\sim P_X, z\sim P_Z} e^{f(x)g(z)} + 1 \approx I_{\text{MINE-f}}, I_{\text{EB}}
\end{aligned}
\tag{10}
$$

It means that within a family of decomposable functions where $T_\theta(X, Z) = f(X)g(Z)$, COCO(X,Z) is an upperbound to the MINE estimates. In addition, the equivalence of COCO(X,Z) = 0 and $I(X, Z) = 0$ seems to suggest a form of mutual information bound. On the other hand, MINE allows the use of non-decomposable $T_\theta(X, Z)$. Existing results on MINE (Poole et al., 2018) seem to suggest that a non-decomposable $T_\theta(X, Z)$ gives superior empirical mutual information estimation performance over a decomposable $T_\theta(X, Z)$. The necessity of non-decomposable $T_\theta(X, Z)$ designs and mutual information lower bounds under decomposable designs of $T_\theta(X, Z)$ may be subjects of further research.

Similar to the MINE estimators, HSIC-based estimators tend to have loose confidence intervals due to the need to bound generalization error of kernels $f$ and $g$ on unseen data points. We expect a cross-validation-based approach like DEMINE to also improve the performance of the HSIC-based estimators.

**Comparison between DEMINE and HSIC on synthetic benchmarks.** We compare Canonical Correlation Analysis (CCA), DEMINE, DEMINE-meta and HSIC for independent testing on our 4 synthetic Gaussian and sine wave benchmarks presented in Section 5. Results for a single random seed is reported for a compact presentation, but we have ran experiments using multiple random seeds and find the result of a single random seed representative enough.

For CCA, we compute p-value using the $\chi^2$ test. For HSIC, we report p-value using a publicly available implementation for a spectral HSIC test (Zhang et al., 2018)[11]. The default kernel is used. Hyperparameters are set to recommended setting when available. For DEMINE and DEMINE-meta, the setup is identical to Section 5. A 2-sided 95% confidence interval is reported, but showing only the lower side.

Experiment results are compiled in Table 3. Statistically significant dependence detections with $p < 0.05$ are bolded. Results show that spectral HSIC requires less data to test dependency for the simple Gaussians dataset. But on the more challenging sine wave dataset, DEMINE-sig and DEMINE-meta-sig perform better. Overall, we find DEMINE more complementary to linear correlations for dependency testing on complex signals. Note that Gaussian kernels are used for spectral HSIC. More complex kernels have potential to improve results.

# B    SANITY CHECK ON STATISTICAL DEPENDENCY TESTING

We performed sanity check of our approach, as well as several statistical dependency testing implementations that we compare against. We run different statistical dependency testing implementations on our 1D Gaussian $\rho = 0.0$, $N = 30$ samples dataset where $X$ and $Z$ are independent. A large number of runs with different random seeds are performed. False positive rate of $p < 0.05$ statistical

---

[11]https://github.com/oxmlcs/kerpy. We also experimented with classic HSIC with gamma approximation Gretton et al. (2005a) https://github.com/amber0309/HSIC and a block HSIC implementation (Zhang et al., 2018) from https://github.com/oxmlcs/kerpy, but find that they both report significantly more than 5% false positives for independent 1D and 20D gaussians $\rho = 0$ at $N = \{30, 100\}$ across 100,000-1,000,000 random seeds, indicating errors in confidence interval calculations.

[12]This is a false positive case for CCA, because for this sine wave data ground truth correlation is 0.

Table 3: Statistical dependency testing comparison between CCA, spectral HSIC and DEMINE algorithms on synthetic datasets. Statistically significant dependency with $p < 0.05$ detections are bolded. See text for detailed experimental setups.

| Problem | Samples | CCA | Spectral HSIC | DEMINE-sig | DEMINE-meta-sig |
|---|---|---|---|---|---|
| 1D Gaussian $\rho = 0.8$ | 30 | **p=0.000** | **p=0.001** | $I \geq$ 0.006-0.017 | $I \geq$ 0.006-0.017 |
| 1D Gaussian $\rho = 0.8$ | 100 | **p=0.000** | **p=0.001** | $I \geq$ 0.006-0.009 | $I \geq$ 0.006-0.009 |
| 1D Gaussian $\rho = 0.8$ | 300 | **p=0.000** | **p=0.001** | $I \geq$ **0.143-0.132** | $I \geq$ **0.146-0.132** |
| 1D Gaussian $\rho = 0.8$ | 1000 | **p=0.000** | **p=0.001** | $I \geq$ **0.278-0.168** | $I \geq$ **0.292-0.168** |
| 1D Gaussian $\rho = 0.8$ | 3000 | **p=0.000** | **p=0.001** | $I \geq$ **0.365-0.146** | $I \geq$ **0.344-0.146** |
| 20D Gaussian $\rho = 0.3$ | 30 | **p=0.015** | **p=0.019** | $I \geq$ -0.003-0.017 | $I \geq$ 0.001-0.017 |
| 20D Gaussian $\rho = 0.3$ | 100 | **p=0.000** | **p=0.001** | $I \geq$ 0.000-0.009 | $I \geq$ 0.001-0.009 |
| 20D Gaussian $\rho = 0.3$ | 300 | **p=0.000** | **p=0.001** | $I \geq$ 0.005-0.005 | $I \geq$ **0.007-0.005** |
| 20D Gaussian $\rho = 0.3$ | 1000 | **p=0.000** | **p=0.001** | $I \geq$ **0.322-0.170** | $I \geq$ **0.376-0.170** |
| 20D Gaussian $\rho = 0.3$ | 3000 | **p=0.000** | **p=0.001** | $I \geq$ **0.632-0.253** | $I \geq$ **0.689-0.253** |
| 20D Gaussian $\rho = 0.0$ | 300 | p=0.624 | p=0.498 | $I \geq$ 0.000-0.005 | $I \geq$ 0.000-0.005 |
| 20D Gaussian $\rho = 0.1$ | 300 | **p=0.000** | **p=0.014** | $I \geq$ 0.000-0.005 | $I \geq$ 0.000-0.005 |
| 20D Gaussian $\rho = 0.2$ | 300 | **p=0.000** | **p=0.001** | $I \geq$ 0.002-0.005 | $I \geq$ 0.003-0.005 |
| 20D Gaussian $\rho = 0.3$ | 300 | **p=0.000** | **p=0.001** | $I \geq$ 0.005-0.005 | $I \geq$ **0.007-0.005** |
| 20D Gaussian $\rho = 0.4$ | 300 | **p=0.000** | **p=0.001** | $I \geq$ **0.191-0.140** | $I \geq$ **0.260-0.140** |
| 20D Gaussian $\rho = 0.5$ | 300 | **p=0.000** | **p=0.001** | $I \geq$ **0.621-0.388** | $I \geq$ **0.815-0.388** |
| Sine wave $a = 8\pi$ | 30 | p=0.826 | p=0.856 | $I \geq$ 0.000-0.005 | $I \geq$ -0.002-0.017 |
| Sine wave $a = 8\pi$ | 100 | p=0.962 | p=0.913 | $I \geq$ 0.006-0.009 | $I \geq$ 0.006-0.009 |
| Sine wave $a = 8\pi$ | 300 | p=0.093 | p=0.498 | $I \geq$ **0.448-0.306** | $I \geq$ **0.511-0.306** |
| Sine wave $a = 8\pi$ | 1000 | **p=0.047**[12] | p=0.111 | $I \geq$ **1.351-0.450** | $I \geq$ **1.316-0.450** |
| Sine wave $a = 8\pi$ | 3000 | p=0.166 | p=0.094 | $I \geq$ **1.711-0.442** | $I \geq$ **1.756-0.442** |

Table 4: Sanity check of several statistical dependency testing implementations. See text for details.

| Approach | False Positive Rate (95% test confidence) | Number of Runs | Note |
|---|---|---|---|
| DEMINE-sig | 0.000 | 200 | |
| Spectral HSIC | 0.040 | 612500 | https://github.com/oxmlcs/kerpy |
| Block HSIC | **0.418** | 612500 | https://github.com/oxmlcs/kerpy |
| HSIC with Gamma Approximation | **0.055** | 1000000 | https://github.com/amber0309/HSIC |

significance was recorded to validate if different implementations actually follow such false positive rates. Correct implementations should have false positive rate lower or equal to 0.05. Results are summarized in Table 4. Statistically significant deviations (under Hoeffding inequality) are marked in bold font. The number of runs for DEMINE is relatively low, but no false positives were found. Low false positive rate of DEMINE might be due to partly the conservative estimation provided by Hoeffding inequality, and partly the generalization gap between train and test splits.

