# OpenReview forum: "A Data-Efficient Mutual Information Neural Estimator for Statistical Dependency Testing"
_ICLR.cc/2020/Conference — Reject_

### Official Review · AnonReviewer1 · 2019-10-23
**Official Blind Review #1**

**Rating:** 6

**Review:**

The paper proposes a neural-network-based estimation of mutual information, following the earlier line of work in [A]. The main focus has been to develop an estimator that can reliably work with small dataset sizes. They first reduce the sample complexity of estimating mutual information by decoupling the network learning problem and the estimation problem by creating a training and validation set and then using the validation set for estimating mutual information. Of course, there is still the problem of learning the network with smaller sized data. For this, they propose the strategy of creating multiple tasks from the same dataset, where the dataset is run through transformations that do not affect mutual information.

I am inclined to accept (weak) the paper for the following reasons:
1. The paper uses some nice ideas to reduce the variance of the MI estimates and to be able to work with smaller dataset sizes. Both splitting data into training and validation and then using task augmentation to make learning robust are pretty nice ideas.
2. The results on the synthetic datasets show that the resulting estimator does have low variance and the estimates are less than or equal to the true MI value, which is consistent with the fact that it is a lower bound estimation.
3. The results on fMRI dataset were interesting and showed that the method gives improvements over baselines were the estimates were made on a smaller sized dataset.

Some improvement suggestions:
1. I don't see why MINE cannot be applied to the fMRI dataset and results be reported. I know that the variance in estimation is large, but it would still be useful to look at the performance of MINE in comparison to DEMINE.
2. There are several errors in the writing - hyperparamters in Abstract, repetition of the word "Section" in fMRI experiment section etc. - which needs to be fixed.

[A] Mutual Information Neural Estimation, ICML 2018

**Experience Assessment:**

I do not know much about this area.

**Review Assessment: Checking Correctness Of Derivations And Theory:**

I assessed the sensibility of the derivations and theory.

**Review Assessment: Checking Correctness Of Experiments:**

I assessed the sensibility of the experiments.

**Review Assessment: Thoroughness In Paper Reading:**

I made a quick assessment of this paper.

---

> ### Author Response · Authors · 2019-11-05
> **First thoughts**
>
> Thank you for the detailed review. We appreciate your comments on the contributions of our work and invaluable suggestions to improving our paper. Before delving into the details and providing a more detailed rebuttal, here are some of our initial thoughts.
>
> Regarding comparison with MINE on fMRI dataset, MINE could be applied to fMRI data for MI analysis. After all, our DEMINE-vr is simply MINE under cross-validation. Our first guess is that MINE will not be able to identify significant dependency in Table.1 due to large confidence intervals, i.e, the rows and cols for MINE will be all 0s. For segment classification, MINE-f-ES uses the same model architecture and training process as DEMINE-vr and uses hyper parameters from DEMINE-vr hyperparameter search, and will provide identical segment classification accuracy as DEMINE-vr.
>
> Thanks again for pointing out the typos, we'll post an updated version shortly.

---

### Official Review · AnonReviewer3 · 2019-10-24
**Official Blind Review #3**

**Rating:** 3

**Review:**

This manuscript studies mutual-information estimation, in particular variational lower bounds, and focuses on reducing their sample complexity. The first contribution is based on adapting the MINE energy-based MI estimator family to out-of-sample testing. MINE involves fitting a very flexible parametric form of the distribution, such as a neural network, to the data to derive a mutual information lower bound. The present work separates the data fitting from the mutual information evaluation to decrease sample complexity, the argument being that the function class is no longer a limiting factor to sample complexity of the mutual information estimation. The second contribution uses meta learning to decrease the sample complexity required to fit the neural network, creating a family of tasks derived from the data with data transformation that do not modify the mutual information. The approaches are demonstrated on synthetic data as well as fMRI data, to detect significant inter-subject dependencies in time-series of neural responses.

There are some very interesting and strong contributions of this manuscript. However, I worry that one of the central theoretical arguments does not seem correct to me. Indeed, the manuscript introduces sample complexity results to justify the benefits of the out-of-sample procedure (th 1), but it seems to me that these give an incomplete picture. Th 1 gives the number of validation samples required to bound error between the mutual information estimate at finite samples and asymptotically for a function T parametrized by \tilda{theta}. This control is of a very different nature from the control established by MIME which controls the error to the actual best possible variational bound. In other terms, the control of th 1 does not control the estimation error of T. This is the reason why it is independent from the function class. The total error in estimating the mutual information must take this error in account, and not only the validation error. Hence the theoretical sample complexities contributed are not comparable to those of MIME.

The meta-learning estimator seems to involve a significant implementation complexity, for instance heuristic switchs between estimation approaches. The danger is that could hard to make reliable in a wide set of applications. It would be more convincing to see more experiments. Along this direction, it is problematic that, in the synthetic examples, the relative  performance of methods changes significantly from experiment to experiment and there does not seem to be a simple way to control that. On the other hand, the MIME-f-ES tends to have a reasonably good failure mode.

I would have been interested in "false detection" experiments: comparing estimator in a variety of problems where the mutual information is zero, but for different marginal distribution. This is particularly important when the application is to test for independence, as in the fMRI experiments.

For hyper-parameter search (using hyperopt), the manuscript should make it explicit what metric is optimized. Is it the data fit of the neural networks? With what specific measure?

With regards to the fMRI experiments, good baselines are missing: DEMINE is compared to Pearson correlation. Additionally, CNNs are not a particularly good architecture for fMRI, as fMRI is not locally translation invariant (see Aydöre ICML 2019 for instance). Finally, it seems that the goal here is to test for independence. In such a situation, there are better strategies for higher-order tests of independence than estimating mutual information, in particular estimator that give a control p-value.

Minor comments:

Alg 1 seems a fairly generic neural-network fitting algorithm. In its current state, I am not sure that it adds a lot to the manuscript.

There are many acronyms that are never defined: MINE, TCPC

**Experience Assessment:**

I have published one or two papers in this area.

**Review Assessment: Checking Correctness Of Derivations And Theory:**

I assessed the sensibility of the derivations and theory.

**Review Assessment: Checking Correctness Of Experiments:**

I assessed the sensibility of the experiments.

**Review Assessment: Thoroughness In Paper Reading:**

I read the paper at least twice and used my best judgement in assessing the paper.

---

> ### Author Response · Authors · 2019-11-05
> **First thoughts**
>
> Thank you for the detailed review. We appreciate your comments on the contributions of our work and the nature of our approach, as well as suggestion of experiments and paper writing. Before delving in and providing more details, we have some initial thoughts about the theoretical issues you brought up.
>
> Regarding theorem 1 and the sample complexity of MINE, we also had discussions on why we think they are comparable or not and discussed that on page 5 in our submission. The tldr is that the MINE sample complexity can not only be seen as 1) bounding best achievable MI estimation but also as 2) bounding distance from estimation to a proven MI lower bound. The former is a quite vacuous bound on generalization and would require advances in learning theory to improve, not MI estimation. Our theorem 1 is trying to improve the latter to enable practical applications. Improving the former to the level of practical use is a noble goal, let us know when you have an answer.
>
> Regarding "false detection" experiments. We really appreciate that you brought up this point. Our synthetic experiments on Gaussians rho=0.0 in Figure 1 do exactly this. Results show that MINE-f and MINE-f-ES estimates very much non-zero MI when there should have been 0 MI. MINE-f bar is not visible due to overshooting out of the chart. DEMINE approaches give estimations closer to 0.  We often get questions about why our estimators give MI numbers lower than MINE and why are we claiming that our estimator is better. But in fact that's exactly because MINE gives false detection but our estimators provably don't.
>
> Hyper-parameter search example. Say we are given 3000 paired (x,y) observations. First divide into 1500 train, 1500 test. Take 1500 train and run Algorithm 1 using 3-fold crossval: use 1000 for (x,y)train and 500 for (x,y)val in each run (3 runs total). Get MI estimation m1,m2,m3 over 3 folds. Compute confidence interval v using Eq.8 using the hyperparameters and 1500 as test set size.  Hyperparameter search DEMINE-vr maximizes mean([m1,m2,m3])-2std([m1,m2,m3]). DEMINE-sig maximizes mean([m1,m2,m3])-v. Will try to make it more clear.
>
> Regarding fMRI experiments, our focus is on demonstrating neural MI estimation and dependency test on fMRI data. We compare with pearson's correlation because it's another widely used technique that can perform both correlation analysis as well as significance test.  We used a simple 1D CNN where convolution happens over the time dimension, not the spatial dimensions. Better architectures, e.g. transformers over time + graph networks over space could improve performance, but not our focus and we leave that to future work.
>
> Higher-order and nonlinear covariance tests may make very appealing comparisons and we are looking into it. A first impression is that our technique is more general and will probably give looser bounds, but may be applicable to a wider range of problems not only ones that have specifically that type of covariance, just like DEMINE vs Pearson's correlation. But at the same time we also have questions on how much additional insight it brings, as it's not an apples to apples comparison, so neither tight or loose estimations diminish the value of both types of approaches.

---

### Official Review · AnonReviewer4 · 2019-11-04
**Official Blind Review #4**

**Rating:** 1

**Review:**

Second update:
We thank the authors for updating their paper.

The work is now improving, and is on the right track for publication at a future conference.  There are a few comments on the new results, and suggestions for further improvement:

* The issue of possible false positives due to sample dependence in fMRI data has again been ignored in the rebuttal. Without careful treatment of this effect, these results are vacuous.

* There is still no comparison with competing nonparametric tests on the fMRI data.

* the results linking the COCO and MINE estimators are interesting. Some statements don't make sense, however, eg. "HSIC-based estimators tend to have loose confidence intervals due to the need to bound generalization error of kernels f and g on unseen data points." First, f and g are functions, not kernels. Second, testing with HSIC or COCO does not require generalisation to unseen data points: this is why testing is an easier problem than regression.

For HSIC testing, I am surprised to read in the footnote that the Gamma approximation report significantly more than 5% errors, especially given the Table 4 results that show the correct level. In any case, I very strongly suggest using a permutation approach to obtain the test threshold for HSIC, which is by far the most robust and reliable method. The Gamma approximation has no statistical guarantees, as stated explicitly in the HSIC testing paper of NeurIPS 2007. Permutation gives a guarantee of the correct level. See the NeurIPS paper for details. Once you have verified the correct false positive rate for the permutation threshold, then you can compute the p-value on the alternative.

While the comparison with HSIC is a helpful one, it is also required to compare with the competing method closest to yours, i.e. the Berrett  and Samworth test. In addition, HSIC is a non-adaptive test, but your test is adaptive, so a fairer comparison would be to a modern adaptive test such as "An Adaptive Test of Independence with Analytic Kernel Embeddings."

* The false positive rate in the sanity check is far below the design level of 0.05. This is as I expected, given the use of the Hoeffding bound. This should be stated clearly in the main text, and not disclosed in the final sentence of the final page of the appendix.


====================

Update: thank you for your rebuttal.

"the necessary and sufficient condition of dependency, is more general and is complementary to other techniques that make stronger assumptions about the data. "

The alternative tests listed are nonparametric. That is to say, unlike Pearson correlation, they do not make specific parametric assumptions on the nature of the dependence. Rather they make generic smoothness assumptions (your test also makes such assumptions by the choice of neural network architecture). Thus, comparison with the prior work in Statistics and Machine Learning is relevant, since these tests have the same aims and scope as your tests.

The cited Berrett and Samworth MI test uses a permutation approach to obtaining the test threshold, not an asymptotic approach  (see the results of Section 4 of that paper).   Several of the other cited tests also use a permutation approach for the test threhsold. These tests are therefore relevant prior work.

" If things change very little from one second to the next, the signals could be very similar and may not really be, intuitively, independent samples and may bias result of the study. However, which independence assumptions to use is not in scope for our paper,  because our fMRI study is trying to show that dependency testing works "

In the work cited by Chwialkowski et al, failure to account for the dependence between samples results in excessive false positives. This is because, for dependent data, the effective sample size is reduced, and the tests must be made more conservative to correct for this effect. It is therefore the case that the fMRI results may be false positives.

Re level:"This proof could be experimentally verified ..."  This should be verified. In particular, Hoeffding can be very loose in practice, which is likely to be observed in experiments.


======

The authors propose a procedure for improving neural mutual information estimates, via a combination of data augmentation and cross validation. They then use these estimates in hypothesis testing, on low dimensional toy datasets and on high dimensional real-world fMRI data.

The improved training procedures for MI estimation are of interest, however the hypothesis testing parts of the paper could still be improved.

In hypothesis testing, it is important to verify that the test has the correct level (false positive rate). This is all the more essential when the estimate has required optimisation over parameters. It is not clear from the presentation that this has been confirmed.

There are a number of prior approaches to testing for multivariate statistical dependence in the machine learning and statistics literature (including a 2017 paper which uses mutual information). A small selection is given below, although a literature search will reveal many more papers. In the absence of citation or comparison with any of the prior work on multivariate statistical dependence testing, the current submission is not suitable for publication.


In statistics:
---------------

https://arxiv.org/abs/1711.06642
Nonparametric independence testing via mutual information
Thomas B. Berrett, Richard J. Samworth
2017


Measuring and testing dependence by correlation of distances
Gábor J. Székely, Maria L. Rizzo, and Nail K. Bakirov
Ann. Statist.
Volume 35, Number 6 (2007), 2769-2794.


Large-scale kernel methods for independence testing
Qinyi ZhangEmail Sarah Filippi, Arthur Gretton, Dino Sejdinovic
Statistics and Computing
January 2018, Volume 28, Issue 1, pp 113–130| Cite as



In machine learning:
---------------------

Multivariate tests of association based on univariate tests
Heller, Ruth and Heller, Yair
Advances in Neural Information Processing Systems 29
2016

A Kernel Statistical Test of Independence
Gretton, Arthur and Fukumizu, Kenji and Choon H. Teo and Song, Le and Sch\"{o}lkopf, Bernhard and Alex J. Smola
Advances in Neural Information Processing Systems 20
2008
http://papers.nips.cc/paper/3201-a-kernel-statistical-test-of-independence.pdf

http://proceedings.mlr.press/v70/jitkrittum17a/jitkrittum17a.pdf
An Adaptive Test of Independence with Analytic Kernel Embeddings
Wittawat Jitkrittum, Zoltán Szabó, Arthur Gretton ; ICML 2017, PMLR 70:1742-1751

Time dependence
----------------

It is also the case that if the variables have time dependence, then appropriate corrections must be made for the test threshold, to avoid excessive false positives. Does the fMRI data exhibit time dependence?  For the case of multivariate statistical dependence testing, such corrections are described e.g. in:

https://papers.nips.cc/paper/5452-a-wild-bootstrap-for-degenerate-kernel-tests.pdf
A Wild Bootstrap for Degenerate Kernel Tests
Kacper Chwialkowski, Dino Sejdinovic, Arthur Gretton
NeurIPS 2014



**Experience Assessment:**

I have published in this field for several years.

**Review Assessment: Checking Correctness Of Derivations And Theory:**

I assessed the sensibility of the derivations and theory.

**Review Assessment: Checking Correctness Of Experiments:**

I assessed the sensibility of the experiments.

**Review Assessment: Thoroughness In Paper Reading:**

I made a quick assessment of this paper.

---

> ### Author Response · Authors · 2019-11-05
> **First thoughts**
>
> Thank you for the detailed review. We appreciate your extremely useful pointers to existing dependency testing techniques, many of which are new to us. Before delving into them, here's our initial thoughts.
>
> First and foremost, we would like to know more details on the reasoning behind the rejection rating. It seems that the criticisms are on citations to other dependency testing approaches and time-dependency of fMRI data. The suggestions are invaluable and we'll gladly include them (work in progress). But comparison with them is not apples to apples , so we are not sure to what extent that adds value to our work, where we already compared to the KSG estimator for MI estimation and Pearson's correlation for dependency testing.
>
> Comparing to other dependency testing approaches, our technique allows the use of arbitrary neural networks and directly tests MI>0, the necessary and sufficient condition of dependency, is more general and is complementary to other techniques that make stronger assumptions about the data. We want to point out that a simple and most widely used dependency testing technique is Pearson's correlation through test of linear correlation (sufficient condition of dependency), which we compared to and show that our technique provides complementary value on sine wave and the fMRI dataset. We are aware that low sample complexity dependency tests exist and can be achieved by making additional assumptions on data, and we discussed that in the conclusion section, but our technique makes less assumptions and is applicable to general datasets which may not satisfy stricter assumptions. We are very interested in the HSIC-based techniques which seem to be popular and we could show complementarity. But at the same time the conclusion will be the same, so we have question on what value does it add for the audience over Pearson's correlation.
>
> Regarding other methods for dependency testing through mutual information, after following the line of work by Barrett et al. 2017, we reached this concise summary http://ims-vilnius2018.com/content/pdf/ivc293.pdf, which explains the smoothness assumptions made to data, as well as the fact that they used an asymptotic variance which holds when a large number of samples is given (page 2 top). It implies that the resulting confidence intervals as well as the test results are asymptotic and not guaranteed, which puts the resulting statistical tests into question. We have to admit that we did not thoroughly understand this line of work because of our background, so please comment if we are wrong about that. Instead, our dependency test does not make assumptions about data. Our lower bound and its confidence interval are not asymptotic. Theorem 1 provides a guaranteed confidence interval for arbitrary number of samples (so do our baselines, MINE and Pearson's r). We compared with the KSG estimator, which also only has asymptotic confidence intervals in literature. In addition, the proof provided in our work is concise and is easy for readers to understand and verify.
>
> Regarding time dependence and test threshold, it's important and thanks for pointing it out! We think that there are two ways time affects dependency listed below.
> 1) First, we assume non-overlapping windows of TRs as the basis for computing number of i.i.d fMRI samples, but did not mention that in our current draft. We'll update and make it clear.
> 2) Second, we can see an argument on whether or not segments of fMRI signals qualify as i.i.d, although not sure if this is a problem for our MI estimation approach. If things change very little from one second to the next, the signals could be very similar and may not really be, intuitively, independent samples and may bias result of the study. However, which independence assumptions to use is not in scope for our paper, because our fMRI study is trying to show that dependency testing works and is complementary to Pearson's correlation, not so much on drawing neuroscience conclusions.  On a side note, it may turn out that which independence assumptions to make is a deeper question that doesn't yet have a clear answer.
>
> Regarding the level of the test, our theorem 1 already provides a proof based on Hoeffding inequality. This proof could be experimentally verified through computing MI on 300 test set samples and see how the estimate changes if there were >1million test set samples, repeat say 1000 times using different random seeds.

---

### Author Response · Authors · 2019-11-15
**Responses to reviewer comments and discussions**

We thank the reviewers for their detailed reading, comments and discussions. We've reviewed the paper, incorporated the reviewers comments and added some of the experiments requested by the reviewers.

Reviewer #4 requested multiple experiments that we believe is out of scope for this work. In this revision we have included results of HSIC, a relevant approach that the reviewer requested, as it is equivalent to testing MI = 0. The goal of this work is not to provide a full benchmark for comparison, which would require an effort that goes beyond this work, and does not currently exist.  We believe that what the reviewer is asking us to do is would be a great survey and benchmarking paper which is not our goal.  We compared against related work, which did not benchmark against all parametric and non-parametric approaches, in addition to the HSIC approach requested as it adds value to the paper. We hope that Reviewer #4 is understanding of the time commitment to fulfill the request to benchmark against the slew of approaches proposed in the review.  We again appreciate the effort referencing them, and we will consider submitting a different paper that would set the stage for benchmarking in this field as it could be of value based on this discussion.

We addressed Reviewer #3's comments on the reasoning behind the sample complexity comparison between MINE and our approach in our earlier reply. In addition, we also added experiments for a sanity check of our confidence interval calculation in results Table 4 in Appendix A.3, which is requested by both Reviewer #4 and Reviewer #3. Although a limited number of repeats is performed due to our available resources, but our result shows a conservative yet conforming 0 false positive rate, with possible reasons discussed in the paper.

Following Reviewer #1's comments, we have improved the writing of our paper. Typos in the public abstract and keywords are still present, because we could not update it. But corrections are already made in the paper.

---

### Decision · Program_Chairs · 2019-12-19

**Decision:**

Reject

**Comment:**

The paper deal with a mutual information based dependency test.

The reviewers have provided extensive and constructive feedback on the paper. The authors have in turn given detailed response withsome new experiments and plans for improvement.

Overall the reviewers are not convinced the paper is ready for publication.